# Protein Intake and Oral Health in Older Adults—A Narrative Review

**DOI:** 10.3390/nu14214478

**Published:** 2022-10-25

**Authors:** Thilini N. Jayasinghe, Sanaa Harrass, Sharon Erdrich, Shalinie King, Joerg Eberhard

**Affiliations:** 1The Charles Perkins Centre, The University of Sydney, Sydney, NSW 2006, Australia; 2School of Dentistry, Faculty of Medicine and Health, The University of Sydney, Sydney, NSW 2006, Australia; 3School of Life Sciences, University of Technology Sydney, Sydney, NSW 2007, Australia; 4School of Pharmacy, Faculty of Medicine and Health, The University of Sydney, Sydney, NSW 2006, Australia

**Keywords:** older adults, protein intake, periodontitis, amino acid composition, oral health

## Abstract

Oral health is vital to general health and well-being for all ages, and as with other chronic conditions, oral health problems increase with age. There is a bi-directional link between nutrition and oral health, in that nutrition affects the health of oral tissues and saliva, and the health of the mouth may affect the foods consumed. Evidence suggests that a healthy diet generally has a positive impact on oral health in older adults. Although studies examining the direct link between oral health and protein intake in older adults are limited, some have explored the relationship via malnutrition, which is also prevalent among older adults. Protein–energy malnutrition (PEM) may be associated with poor oral health, dental caries, enamel hypoplasia, and salivary gland atrophy. This narrative review presents the theoretical evidence on the impact of dietary protein and amino acid composition on oral health, and their combined impact on overall health in older adults.

## 1. Introduction

Oral health encompasses the condition of a person’s teeth, gums, oral secretions, jaw bones, and facial muscles [1], and is a key indicator of overall health, well-being, and quality of life [2,3]. While oral health problems increase with age, adults over the age of 65 have more such problems than the rest of the population [4]. These problems include tooth loss, tooth decay, periodontal (gum) disease, dry mouth, and oral cancer, all of which can significantly impact general health [5], and may be the direct result of suboptimal care of the teeth and mouth [6].

Periodontitis (gum disease) is highly prevalent in older adults. Around 42% of dentate adults over 30 years old have periodontitis, which increases to >60% in adults over 65 years [7,8]. Periodontitis is an inflammatory disease characterised by the destruction of connective tissues and alveolar bone surrounding the teeth, and is a major cause of tooth loss. Periodontitis is associated with chronic diseases, particularly hyperglycemic states associated with poorly controlled diabetes mellitus [9], cardiovascular disease [10] and chronic kidney disease [11]. Factors associated with ageing also increase the risk of dental caries (tooth decay) in older adults due to an ecological imbalance in oral biofilm, leading to the demineralisation of teeth [12]. A recent systematic review reported that 50% or more of older adults had untreated dental caries [13]. Evidence from both association and controlled clinical depletion studies show that periodontitis and dental caries may be influenced by inherited risk factors, as well as those acquired over a lifetime, including poor oral health and sub-optimal dietary patterns [14,15,16,17,18]. A consensus report [14] concluded that the role of genetic factors in the development of periodontal diseases and caries was moderately strong, possibly contributing up to 50% of the risk. Much of this inherited risk is associated with specific genes associated with immune function (Fc gamma receptor, IL10) and the vitamin D receptor [14].

Oral structures, such as teeth, tongue, and salivary glands, play significant roles in maintaining a person’s oral health. At least twenty teeth are considered necessary for a functional dentition [19]. The tongue is equipped with muscles, nerves and hormones, together regulating taste perception and satiety, and helping mastication and swallowing food [20]. Salivary glands produce approximately 0.5–1.5 L of saliva daily [21]. Saliva contains specific antimicrobial proteins such as lysozyme, lactoferrin, peroxidase enzymes, histatin and proline-rich proteins, and other substances, such as mucins, glycoproteins, fibronectin, beta-macroglobulin, lysozyme, and secretory-IgA, that clump bacteria [22,23]. Saliva also protects the oral and peri-oral tissues via lubrication, antimicrobial and cleansing activity, buffering (neutralising) acid production, controlling plaque pH with bicarbonate, and enhancing chewing, swallowing and initiation of digestion [24]. The structure of salivary glands and the flow and composition of saliva tend to change with ageing and age-associated diseases (such as Sjögren’s syndrome) [25]. When saliva flow is reduced, and the pH is low (due to reduced buffering capacity), oral health problems such as dental caries and oral infections may develop due to acidic demineralisation of tooth structure and irritation of oral mucosal surfaces [26]. Oral diseases are detrimental to masticatory function, which is a crucial first step for processing food in the mouth. Mastication and subsequent nutrition acquisition are essential determinants of food choices and subsequent nutritional and health status, respectively [27].

Nutrition and oral health are intricately linked. Proper nutrition is essential for establishing good oral health in pre- and post-eruptive phases. Proteins and vitamins A, C and D, as well as calcium, phosphorous and fluoride, are nutrients essential for the development and maintenance of teeth [28]. In addition, the collagen in dentine is dependent on vitamin C for normal synthesis, and keratin incorporation into the enamel requires vitamin A. Daily intake of a variety of nutrient-dense food, rich in the above-mentioned nutrients, promote healthy teeth and gums.

Oral health is vital for ensuring proper dietary intake at any age. Older adults have increased risk of inadequate nourishment due to reduced chewing function because of tooth loss, pain or discomfort (e.g., from poorly fitting dentures), and impaired cognitive function [27,29,30]. For instance, edentulism can reduce the functionality of the mouth, making chewing and swallowing more challenging, thus compromising nutrition, contributing to low body mass index (BMI) [31]. Compromised nutrition due to altered food choices results in individuals selecting soft, easy-to-chew foods, that are often lower in fibre, protein and iron, amongst other nutrients [32]. Importantly, poor nutritional status is both a cause and consequence of poor oral health among older adults [33]. This combination of poor oral health and malnutrition can lead to lasting physical and psychological disabilities, reducing the quality of life of older adults.

It is essential to understand that appropriate nutrition and improved oral health outcomes go hand in hand. There are several expert opinions and multiple sources of evidence on the Recommended Dietary Allowance (RDA) of protein required for the prevention of malnutrition in older adults [34,35]. Protein is a crucial nutrient for older adults, especially as it is imperative for muscle health, maintaining energy balance, weight management, bone mineralisation, and cardiovascular function [36].

Proteins are made of various amino acids (AAs), including non-essential, essential (must be obtained from the diet), and branch-chain AAs (BCAAs) [37]. Protein plays a vital role in oral health as a building block for bone and the periodontium, including its role in tissue repair [38,39]. Protein is also crucial for preventing sarcopenia [40], maintaining normal immune function, and supporting wound healing. Higher protein intake is associated with improved periodontal healing [7]. The evidence about the impact of protein containing oral nutritional supplements (ONS) on the nutritional status of older adults is inconclusive. For instance, the results of a recent randomised control trial showed that daily intake of a nutritionally complete ONS powder improved nutritional outcomes of free-living adults at risk of malnutrition [41]. In contrast, a meta-analysis showed little evidence of ONS reducing malnutrition or its associated adverse outcomes in frail older adults [42].

Previous work has shown that there are associations between low protein intake and malnutrition risk [29,43,44,45], and also higher prevalence of oral health problems in older adults [19,46,47,48]. However, there is insufficient evidence to link protein intake directly to oral health without being mediated by malnutrition. We hypothesise that adequate dietary protein intake is associated with good oral health in older adults. Hence, in this narrative review, we aim to present the current evidence supporting the association between dietary protein intake and oral health. New studies and novel concepts in nutrition research are necessary to examine whether there is a causal relationship between protein intake and oral health in older adults. Once established, future studies are needed to identify the adequate quantity of protein needed for optimal oral health in this specific population.

There are local and other factors affecting low food intake in older adults. Local influences include loss of appetite, loose teeth, edentulism, jaw muscle weakness and dry mouth. Other factors such as cognitive decline, difficulty performing daily life activities (ADLs), multimorbidity, sensory changes and social isolation contribute to low food intake and associated macro- and micronutrient deficiencies [49,50,51,52]. There is a two-way relationship between low food intake and oral health problems, such as tooth loss, tooth decay and periodontitis [38]. Additionally, an unhealthy diet and hyperglycemia also lead to tooth loss, dental caries and periodontal diseases [16,53,54]. Low protein and energy intake caused by inadequate food intake directly leads to protein–energy malnutrition (PEM) in older adults [55]. There is also a bidirectional indirect effect between PEM and oral health issues [56]. However, there is a lack of evidence for a direct relationship between protein intake and oral health.

## 2. Methods

We performed a search of the literature on the PubMed, Web of Science and SciVerse Scopus^®^ databases in June 2022. There were no restrictions on publication dates and the search terms included “older adults or elderly”, “malnutrition or low protein intake”, “oral or dental health issues”. Titles and abstracts of the articles were reviewed to identify scientific publications relevant to our aim. Specifically, articles that were related to associations between protein intake/malnutrition and oral health issues in older adults were gathered and reviewed to determine suitability for inclusion.

## 3. Factors Affecting Optimal Dietary Intake in Older Adults

### 3.1. Local Factors

A gradual decrease in appetite is common among older adults. Ageing affects the endocrine system, including hormones involved in controlling appetite and food intake [57]. Increased activity of cholecystokinin, leptin and various cytokines, and reduced activity of ghrelin [58], make older adults more prone to becoming anorexic and undernourished [59].

Tooth loss increases with age. Over 45% of adults over 75 years of age do not have functional dentition [60]. They often have dentures or may be edentulous (no natural teeth), compromising the ability to chew and swallow food effectively. Tooth loss is an important determinant of chewing function and is associated with poor diet quality [61]. Chewing with dentures is at least 30% to 40% less efficient than with natural teeth [62]. Understandably, these problems may reduce overall dietary intake and alter food choices in favour of a diet that is more comfortable to chew [63].

Dry mouth (xerostomia) is common in older adults due to disease, lower salivary flow rate, or as a side effect of pharmaceutical agents influencing the properties of saliva [25]. Reduced saliva flow in xerostomia increases susceptibility to dental decay and to fungal infections in the mouth [64], alters the sense of taste, and is associated with intolerance of dentures as well as problems related to chewing and swallowing [65,66]. Additionally, the neutral pH of saliva may not be maintained, resulting in the growth of caries-associated microorganisms that prefer an acidic environment [67]. Older adults appear to modify food choices or avoid swallowing hard food in response to perceived dry mouth. This results in inadequate nutrient intake [68].

Polypharmacy is common in older adults due to the increase in disease conditions such as diabetes mellitus, dyslipidaemia, hypertension, heart disease, sleep disturbances and joint pain [69]. Many of the medications taken for these conditions may compromise salivary secretion, either through direct inhibition of secretory mechanisms, or indirectly through altered tissue hydration [70], leading to xerostomia [66]. Additionally, the long-term use of multiple drugs may also cause loss of appetite, reduction in food intake and compromise the absorption of essential vitamins and minerals [71,72]. A recent systematic review reported a statistically significant association between polypharmacy and malnutrition in older adults [73].

### 3.2. Other Factors

Sarcopenia—loss of skeletal muscle mass, strength, and function—is common among older adults, due to age-related changes in physical activity, hormone production, and reduced nutrient intake [29,74]. In turn, sarcopenia can affect swallowing, leading to “sarcopenic dysphagia”, thereby compromising nutrient intake [75]. Furthermore, impairment of sensory performance among older adults has been identified as one of the potential underlying causes for the loss of appetite and inadequate energy intake [76]. This can be exacerbated by comorbidities via the disease/s or the resultant medical treatments.

Cognitive impairment is an age-related problem, prevalent in more than 40% of older adults [77]. Cognitive impairment and poor oral health are also inter-related [78]. For instance, the rejection of care, aggression, and agitation of people with dementia hinders the performance of daily oral care [79]. Additionally, cognitive decline and weak muscle function result in reduced chewing efficacy [80], less frequent tooth brushing and poor oral hygiene, increasing the risk of dental caries and tooth loss [78]. A recent review showed that more missing teeth and higher average probing pocket depths are risk factors for cognitive decline [81]. It has also been demonstrated that cognitive impairment was associated with poor oral health and poorer activities of daily living (ADL) functioning among adults aged over 65 [82]. All the above factors may affect food intake adversely [83] and the subsequent nutritional deficiencies lead to more oral health issues. A healthy diet, high in vegetables, soy products, fruit, and fish, has been shown to benefit cognitive function [84,85,86]. Thus, improving cognitive function would logically have positive effects on both nutrient intake and oral hygiene. Furthermore, higher soy protein intake in rats resulted in increased bone mineral density (BMD) under nonenergy deficiency [87]. Increased BMD may be an indicator of good oral health status in humans [88,89].

## 4. Effect of Various Dietary Protein Sources on Oral Health

Protein is a vital macronutrient in a well-balanced diet, and is essential for growth, muscle strength and function, immune function, wound healing, and overall tissue homeostasis. In addition to general health, dietary proteins play a vital role in good oral health [90].

Not all protein is created equal. Dietary protein comes from non-animal (plants) and animal (meats, eggs, milk) sources. The quality of a protein is determined by its biological value (ratio of essential to non-essential amino acids), protein efficiency ratio (the ability of a protein to support growth), and net protein utilisation (the percentage of amino acids converted to tissue protein versus the amino acids digested). Furthermore, other nutrients in protein-rich foods (especially animal protein), such as calcium and vitamin D, have a beneficial effect on tooth retention in older adults [91]. Importantly, the calcium and phosphorus inherent in dairy foods, such as cheese and milk, help protect teeth against demineralisation by preventing the pH in the mouth from falling below 5.5, thereby reducing the risk of dental decay [90,92]. Additionally, there is an inverse association between the consumption of milk and dairy foods and the prevalence of periodontitis in the adult population [93]. While the exact mechanism behind this association has not yet been revealed yet, researchers postulate that lactic acid in fermented dairy products inhibits the growth of periodontal pathogens by decreasing oral pH [94].

Milk is an excellent protein food, providing essential amino acids and organic nitrogen for humans of all ages—allergic responses and lactose intolerance aside. Data from a rat study indicates that the whey portion of milk protein increases bone collagen, enhances bone strength, and prevents alveolar bone loss by increasing hydroxyproline, which can strengthen the coherence of bone [95,96]. In addition to the predominant milk proteins, casein and whey, there are also minor milk proteins and bioactive peptides, such as lactoferrin and transferrin. Several in vitro, in situ, and in vivo studies have shown that these bioactive dairy peptides reduce the risk of dental decay [97]. For example, salivary lactoferrin contributes to oral antimicrobial defences by inhibiting the growth of bacteria associated with periodontal disease and modulating the associated inflammatory processes [98]. Moreover, casein phosphopeptides–amorphous calcium phosphate (CPP-ACP) is a bioactive agent present in milk that is formulated from casein phosphopeptides (CPP) and amorphous calcium phosphate (ACP). CPP in milk is capable of stabilising calcium phosphate and increasing the calcium phosphate content in dental plaques [99]. Additionally, the incorporation of CPP-ACP into oral care products has been shown to prevent the formation of biofilm by *Streptococcus mutans*, a common cariogenic bacterium in the oral cavity involved in plaque formation [100]. This prevents *S. mutans* from adhering to the tooth surface, thus reducing the risk for dental caries [100]. However, this effect from the ingestion of typical dietary intakes of dairy products has, to date, not been demonstrated.

### Dietary Amino Acid Composition and Its Effect on Oral Health

Multiple AAs are linked by peptide bonds to form a protein. Twenty-one AAs build up the proteins found in humans, of which nine must be obtained from the diet—these are the essential AAs [101].

It is important to consider the AA composition in proteins and how these affect oral health. AAs have a range of impacts on oral tissues; most are beneficial and act by reducing bacterial colonisation of oral tissues, modulating the inflammatory response, which reduces gingivitis and mucositis, reducing the risk of dental decay by enhancing the properties of saliva in neutralising acids or mineral homeostasis, and impacting the immune system to promote the phagocytosis of bacteria. Table 1 details some of the known roles AAs play in oral health.

L-arginine, for example, inhibits bacterial coaggregation in the human oral cavity and stops plaque formation [102]. Kolderman’s study also demonstrated that L-arginine monohydrochloride moderates multi-species oral biofilm development and community composition and enhances the activity of cetylpyridinium chloride, an antimicrobial compound. In adolescents aged 12–15 years, lower levels of histidine (a non-essential AA) appears to increase the risk of dental caries [103].

Valine, leucine, and isoleucine are BCAAs, essential for building muscle, protecting against muscle loss during exercise, and can be converted into energy. Research has indicated a negative impact of imbalanced dietary BCAAs on health and ageing [104], yet their effect on oral health in older adults has been barely explored.

**Table 1 nutrients-14-04478-t001:** Effect of amino acids on oral health.

Amino Acid	Effect on Oral Health	Material	References
Alanine	Alanine and histidine form citrulline. A higher concentration of citrulline in saliva is correlated with periodontitis.	Human	[105]
Arginine	Arginine improves calcium absorption by the formation of soluble complexes with calcium that maintain calcium in an absorbent form, which is important for enamel maturation. Higher concentration saliva in Stage III Grade C generalised periodontitis.	Human	[105]
L-Arginine	L-Arginine monohydrochloride in saliva inhibits bacterial coaggregation in the oral cavity by decreasing the viscosity of extracellular polymeric substances produced by bacteria and altering cellular metabolism resulting in biofilm dispersion and reducing antibiotic tolerance.	Human	[102]
Aspartic acid	Adult age estimation is based on aspartic acid racemisation in dentine.	Human	[106]
Cysteine	Toxic to oral Streptococci through inhibiting an enzymatic step in the valine-leucine biosynthetic pathway.	Human	[107]
	Reduces bacterial biofilm adherence and biofilm biomass.	A multi-species plaque-derived biofilm model	[108]
*N*-Acetyl-L-cysteine (from L-cysteine)	Reduces pain and hypersensitivity of teeth. Protects gingivae from white lesions and oral mucosal inflammation after using bleaching agents.	Human	[109]
	As mouthwash, it treats and prevents gingivitis	Human	[110]
Glutamic acid	Higher in Stage III Grade C generalised periodontitis.	Human	[105]
Glutamine	Topical administration to patients receiving stomatoxic chemotherapy resulted in 20% decrease in moderate and severe oral mucositis.	Human	[111]
Glycine	Glycine supplement reduced dental caries development by 65.7% through the changes in the fatty acid composition of the tooth and a reduction in growth rate (no effect on the retention of either calcium or phosphorus by dietary glycine).	Rodent (rat)	[112]
	Glycine is an integral part of collagen that is an intrinsic component of the tooth structure. Reduced level of saliva glycine has been associated with collagen degradation. Hence, higher salivary glycine has been associated with reduced risk of dental caries and periodontitis through reduced collagen degradation and decreased collagenase activity, leading to less inflammation in gingiva.	Human	[113,114]
Histidine ^†^	Reduces the risk of dental caries. Lack of histidine and its derivatives in saliva results in chelation, i.e., formation of metal complexes with amino acids, leading to initial lesion and secondary to destruction of the organic matrix by the action of proteolytic bacteria.	Human	[103]
Isoleucine ^†^	Found in carious dentine	Human	[115]
Leucine ^†^	Repaired carious enamel.	Human	[116]
	Leucine-rich amelogenin peptide regulates receptor activator of NF-kappa B ligand (RANKL) expression in cementoblast/periodontal ligament cells.	Rodent (mouse)	[117]
Lysine ^†^	Important for the integrity of dentally attached epithelium to act as a barrier to microbial products.	Lysine decarboxylas extracted on *Eikenella corrodens* bacterial cell surface	[118]
Methionine ^†^	Methionine reduces the adverse effect of fluorides on soft tissue, and this has been found to be optimal for the prevention of the adverse effects of chronic fluoride intoxication together with vitamin E in drinking water.	Rodent (rat)	[119]
Phenylalanine ^†^	May inhibit dental caries development. In bacteria, phenylalanine is converted to phenylpropionate or phenylacetate, resulting in alkali environment which is an essential factor in maintaining plaque pH homeostasis.	Human	[120]
Proline	Salivary proline-rich glycoprotein regulates the oral calcium homeostasis by controlling the supersaturated state of saliva with respect to calcium phosphate salts, countering the plaque acidity, formation of dental pellicle, and influencing the composition of plaque.	Human	[121]
	Moreover, this prevents the adherence of oral microorganisms inhibiting their growth and neutralises acids from biofilms protecting from dental caries.	Human	[122]
Serine and threonine ^†^	Interact with host cytoplasmic phosphoproteins, facilitating internalisation of bacteria.	Primary cultures of human gingival epithelial cells	[123,124]
Tryptophan ^†^	Tryptophan metabolites generated from oral supplementation of tryptophan promote regulatory T-cell (Treg) differentiation and suppress proinflammatory T-helper cell (Th)1 and Th17 phenotypes.	Rodent (mice)	[125]
	Higher saliva tryptophan level was observed in Stage III Grade B generalised periodontitis.	Human	[105]
Tyrosine	Potential biomarker of oral lichen planus (lower levels). Tyrosine is suggested to be involved in the antioxidative defence.	Human	[126]
Valine ^†^	Detected in sound dentine compared to carious dentine.	Human	[115]
Homocysteine ^‡^	Associated with high narrow palate, mandibular prognathia (protruding lower jaw), crowding and early eruption of teeth and short dental roots.	Human	[127]

^†^ Essential amino acids, ^‡^ is an AA but not present in the diet and not included in 21AAs.

## 5. Dietary Protein Intake in Older Adults

The recommended dietary allowance (RDA) of protein by the US Food and Nutrition Board for older adults is 0.8 g/kg body weight/day [36]. However, this is well below the ESPEN (European Society for Clinical Nutrition and Metabolism) and the PROT-AGE study group-advocated protein intake of 1.0–1.2 g/kg of body weight per day (g/kg/day) or higher [34,35], reflecting a value at the lowest end of the acceptable macronutrient distribution range of 10–35% of daily calories as protein [36]. Australian guidelines recommend 0.75 g/kg (women)–0.84 g/kg (men) for adults up to 70 years, and 0.94 g/kg (women)–1.07 (men) for >70 year olds [128]. Where protein intake is above the RDA of 0.8 g/kg/day in healthy older adults (52–75 years), whole-body net protein balance has been shown to be greater [129]. It is estimated that about 46% of older adults do not meet the protein intake recommendation [130], and a lack of teeth directly reduces animal protein intake in this age group [37]. Additionally, the protein intake of very old adults (aged ≥ 85 years) were significantly below 0.8 g/kg/day in the Newcastle 85+ Study [131]. Logically, total protein intake and differing AA composition in proteins may have different effects on the oral health of older adults. Significantly, older adults have anabolic resistance to AA intake compared to younger individuals. Therefore, older adults may in fact require higher levels of dietary protein (or essential AA) consumption [132]. Ultimately, insufficient protein and/or energy intake may lead to malnutrition [133].

## 6. Protein–Energy Malnutrition and Oral Health in Older Adults

PEM is prevalent among older adults [134]. It occurs when the body does not receive or absorb enough protein for its physiological metabolism [135]. When PEM occurs in early childhood, it affects the developing immune system, reducing the ability to respond to periodontal pathogens in later life [92]. Anorexia, also common in older adults, contributes to malnutrition due to links to the neurotransmitters and hormones that affect the central feeding drive and the peripheral satiation system [59]. Fluctuating hormones, reduced dentition and some medications decrease the appetite, leading to dietary alterations [136]. These problems are exaggerated due to changes in the absorption and metabolism of essential nutrients that occurs with advancing age.

Malnutrition affects oral health, and poor oral health, in turn, may lead to malnutrition [137]. Research on oral health determinants of community-dwelling older adults identified xerostomia, loss of teeth, and toothache while chewing as determinants of incident malnutrition in community-dwelling older adults [138]. Adverse effects of malnutrition on oral structures include: reduced tooth size, increased enamel solubility, and salivary gland dysfunction [137,139]. There are several mechanisms through which malnutrition is associated with dental caries: (1) enamel hypomineralisation and hypoplasia; (2) under-functioning salivary glands, with reduced flow and changes in composition, affecting buffering capacity and antimicrobial components [23]; and (3) reduced capacity for tissue healing and resisting microbial biofilms [137]. Additionally, it is the lower protein intake that may result in mTORC1 (mammalian target of rapamycin complex 1) failing to activate S6K1 (Ribosomal protein S6 kinase beta-1), leading to less muscle protein synthesis and a possible decrease in chewing function [140]. However, there is a paucity of data on the impact of AA composition and protein intake on oral health.

Hidden hunger, characterised by a deficiency in essential micronutrients, is prevalent in older adults [141]. Therefore, low protein intake, in combination with micronutrient deficiencies, (e.g., vitamin A, zinc and iron), may lead to progression of oral disease including tooth decay [142]. Figure 1 demonstrates the links between low food intake and oral health in older adults.

## 7. How Does Inadequate Protein Intake Affect Saliva and Oral Health?

Saliva secretion is part of good oral health, and a severe reduction in saliva production causes deterioration in oral health [143]. Buffering and antimicrobial activities of saliva are mediated by the carbonate–bicarbonate system, phosphate and antimicrobial proteins in saliva [23]. Specifically, histatins and acidic proline-rich proteins are only found in saliva [22]. Interestingly, histatins have antimicrobial activities against oral pathogens such as *S. mutans* and *Porphyromonas gingivalis,* the main microorganisms implicated in dental caries and periodontitis [22,144]. Furthermore, proline-rich proteins play a role in mineral homeostasis and toxin neutralisation [22].

Dietary patterns determine changes in saliva composition [145]. For example, a liquid diet may significantly alter saliva’s protein composition by reducing protein and electrolyte concentrations [146]. Diets high in vegetables, fruit and high-protein food are associated with changes in the salivary proteome [145]. As stated in the introduction, saliva is composed of numerous defence proteins involved in both innate and acquired immunity [147]. However, the influence of dietary protein components alone on saliva composition and its immune properties, and how this affects oral health, is not yet known.

### 7.1. Protein Intake and Tooth Decay

Factors associated with ageing also increase the risk of dental caries in older adults due to an ecological imbalance in oral biofilm, leading to the demineralisation of teeth [12]. This is shown by the Centre for Disease Control and Prevention (CDC), which reported that 96% of adults aged 65 years or older have had at least one cavity, whilst one in five suffer from untreated tooth decay [6]. Dietary patterns, specifically foods high in sugars, are associated with the prevalence of dental caries [148], as bacteria living in plaque ferment carbohydrates and sugars in food and produce acids, which erode tooth enamel and cause tooth decay [149].

Moreover, the gingival tissue in humans has rapid turnover rates in the human body (4–6 days), and proteins play a key role in this turnover, as discussed in a narrative review [16]. We have not identified any research examining the effect of a protein-rich diet on tooth decay; however, one could argue that a diet high in protein and fat will typically be lower in carbohydrates and sugar, thus making the teeth more resistant to decay. Moreover, a diet rich in protein and fat and low in sugar may also improve the dentinal fluid flow (i.e., from inside to outside of the tooth), ensuring the teeth are more resistant to decay [150]. Furthermore, protein deficiency—rather than caloric malnutrition—leads to increased tooth decay in vitamin-A-deficient rats [151] through increased enamel solubility [152].

### 7.2. Protein Intake and Periodontitis

Periodontitis is highly prevalent in older adults and its prevalence increases with age [7,8]. Periodontitis is an inflammatory disease characterised by the destruction of periodontal tissues and alveolar bone surrounding the teeth. Protein intake in excess of 1 g/kg/day is associated with a reduction of periodontitis [7], and dietary calcium, particularly from milk, may also be protective [153]. A study in a rat model of periodontitis found that higher intake of milk protein (1% versus 0.2%) was protective against alveolar bone loss in periodontitis [96], but to date this has not been explored in humans.

### 7.3. Protein Intake and Tooth Loss

According to the CDC, 13% of Americans aged 65 to 74 years old have no teeth. This doubles in those over the age of 75 [6], resulting in compromised nutritional intake [27]. Data from the British Regional Heart Study showed that a persistent low-protein, poor-quality diet was associated with a higher risk of tooth loss among older individuals [37,38]. Conversely, excellent oral function (optimal number of teeth and ability to chew) is associated with higher consumption of protein and vegetables [154].

### 7.4. Protein Intake and Oral Mucosal Lesions and Oral Cancers

Additionally, oral mucosal lesions are widely prevalent in the geriatric population [155,156]. Systemic diseases, nutritional disorders, medication side effects or ill-fitting dentures have been identified as causes for those lesions [157]. Moreover, the American Cancer Society (ACS) reports that the number of new cases of oral cavities or oropharyngeal cancer in 2022 is about 54,000, and the average age of most people diagnosed with these cancers is 63 [158].

## 8. Excessive Protein Intake and Oral Health

While the impact of insufficient dietary protein on oral health is important, noteworthy is that protein excess can result in low-grade metabolic acidosis (LGMA). In this state, the acidic residue of proteins causes a slight decrease in blood pH, which remains within the normal range. Compensation occurs as the kidneys excrete some of the excess acid [159]. Outright metabolic acidosis has been demonstrated to inhibit dentine formation and promote both the onset and progression of dental caries [160], an issue well understood in chronic renal disease [161].

The potential renal acid load (PRAL) is an estimation of the net acidic contribution of food, where protein and phosphorus increase the acid potential, which is compensated for by the calcium, magnesium, and potassium content of the diet [162,163]. Thus, LGMA may become more problematic when dietary intake of these buffering minerals is low. This is compounded by the increased urinary losses of calcium and phosphorus that occurs in this slightly acidotic state.

Where LGMA is chronically occurring, bone resorption increases and bone mineral density decreases, thus negatively impacting bone and oral health [159]. Interestingly, in addition to supplementation with these minerals, exercise may further counter the risk of LGMA where protein intake is high [164]. This merits consideration when postulating increasing protein intake above recommended intakes in the sedentary older adult population.

## 9. Discussion

Oral diseases are highly prevalent world-wide, especially among the older age groups. Diet, nutrition, and systemic well-being are fundamental in maintaining oral health and general health. Multidirectional synergism exists between diet, nutrition, and oral health. Poor nutritional status can adversely affect oral health, and poor oral health impacts food choices, limiting dietary intake, and subsequently leading to malnutrition. This review has highlighted the importance of dietary protein as a contributor to better oral health, potentially by reducing the risk for decay and periodontitis and improving saliva function. Moreover, it also highlights the bidirectional relationship between oral health and malnutrition.

A healthy, balanced diet consists of a variety of foods that provide sufficient macro- and micronutrients. Total daily calories from protein should ideally provide between the mid-to-higher end of the acceptable macronutrient distribution range of 10–35% of total daily calories in order to maintain healthy ageing, and positively influence both general and oral health. While decreased protein intake is a known risk factor for frailty and sarcopenia in older adults [45], few studies have been conducted on protein intake and oral health in older adults.

The number, structure and distribution of teeth, and quantity and quality of saliva are key local determinants of oral health. Many other systemic factors including polypharmacy, physical and cognitive function are also important factors affecting oral health in the older adults. There are significant challenges for older individuals to achieve proper oral health. The capacity to perform oral health-related activities such as tooth brushing and denture care declines with advanced ageing and cognitive decline, affecting oral hygiene. Poor oral hygiene compromised diet and dry mouth all contribute to deterioration of oral health.

There is evidence to support the importance of a healthy diet in maintaining good oral health in older adults [33,165,166]. However, due to the challenges outlined above, many older adults are at risk of compromised nutrition. *Protein* is a critical macronutrient essential for oral health. Good oral function, which includes an optimal number of teeth and chewing ability, is associated with higher protein consumption [33]. Unlike many other nutrients, such as calcium and vitamin D, that have higher requirements with ageing, the RDA for protein does not increase with age during adulthood in many countries, yet the actual protein intake of older adults is below the current recommendation of 0.8 g/kg/day by the Food and Nutrition Board [167]. Thus, many older adults consume insufficient dietary protein to maintain physical function and optimal health [132]. Data from Morais et al. [40] indicated that protein intake of 1.0–1.3 g/kg/day is required to maintain nitrogen balance in healthy older adults, likely due to lower energy intakes and impaired insulin action during feeding compared with young people. Hence, new research is needed to determine the protein needs and optimal patterns of protein intake for older adults.

To date, there appears to be limited research demonstrating the effect of protein intake on the oral health of older adults. However, there are assumptions that diets low in sugar or carbohydrate tend to be higher in protein or fat and are therefore protective against oral diseases, specifically dental decay. Together with decay, periodontitis is the leading cause of tooth loss in adults. Diet is considered a key modifiable factor in periodontitis, impacting risk via systemic and/or local effects [168]. As evidenced in early rodent studies, protein-depleted diets caused breakdown of periodontal ligaments, degeneration of gingival tissues, and resorption of alveolar bone [169,170]. A cross-sectional study conducted in Denmark suggested an inverse relationship between casein and whey protein intake and periodontitis in adults [171]. Additionally, a randomised controlled study demonstrated that a higher-fat, semi-vegetarian diet is protective against clinical parameters of periodontitis in healthy adults aged 65–75 years [172]. In a cross-sectional study, vegetarians aged ~40 years had smaller pocket depth, and less bleeding on probing, compared to non-vegetarians [173]. However, it is unclear whether dietary protein content was responsible for the protection against periodontitis observed in these studies. This highlights further gaps in the research on the impact of the quantity and source of protein (plant vs. animal) on the oral health of older adults.

The main purpose of this narrative review was to deepen understanding in the broader area of oral health and protein intake in older adults, identifying and summarising the available evidence. We also explored the existing debates regarding optimal protein intake of older adults, appraising previous studies conducted on this specific area to identify knowledge gaps.

As in any narrative review, we had no explicit criteria for article selection. We did not conduct a formal assessment of the quality or risk of bias of included studies, which can be considered a limitation of this review. The authors acknowledge that a systematic review, as a gold-standard summary of evidence, is best to guide clinical decision-making and inform policy development. However, there is insufficient evidence from interventional and cohort studies in this area to inform such a review, or to identify cause-and-effect relationships. Bringing attention to the need for investigations related to oral health and protein intake in older adults is an important step in raising awareness of the need for more evidence on this topic.

In summary, the older adult population are at increased risk of poor oral health, leading to food avoidance, which may explain the increased rates of protein–energy malnutrition. Hence, there should be an emphasis on the need for new strategies by service providers at various care levels aiming at ensuring adequate protein intake to bridge the gap between diet generally and good oral health practices in older adults. For example: (1) Dentists: refer patients with dry mouth and periodontitis to a dietitian for dietary assessment; (2) dietitians: perform a routine diet assessment to review protein intake, include questions about any chewing difficulties and raise awareness of the need for optimal protein intake; (3) clinicians and dietitians: develop educational resources for service providers (e.g., aged care providers) aimed at encouraging clients to consume adequate protein; and (4) food production and processing industry: develop products targeted for older adult populations with improved quality (e.g., texture) and quantity of protein, targeted for older adult populations. Moreover, interdisciplinary teams of general practitioners, dentists, nurses, and nutritionists/dietitians can help by encouraging healthier eating patterns and appropriate nutrition. By ensuring that those at risk maintain good oral health, improvements in long-term health and enhanced quality of life may thus ensue. Further research is warranted to determine the optimal protein needs for older adults to maintain good oral health.

## 10. Conclusions

Older adults experience both oral health issues and malnutrition, which have a coadjutant effect on each other. Evidence suggests that the effect of protein intake on oral health may be mediated through malnutrition. There is a tendency for lower protein intake among older adults, yet the impact of this on oral health is unexplored. Examination of the impact of inadequate protein intake on oral health in this demographic is required. Evaluation of associations would inform guidelines for optimal protein requirements for ensuring good oral health in older adults.

## Figures and Tables

**Figure 1 nutrients-14-04478-f001:**
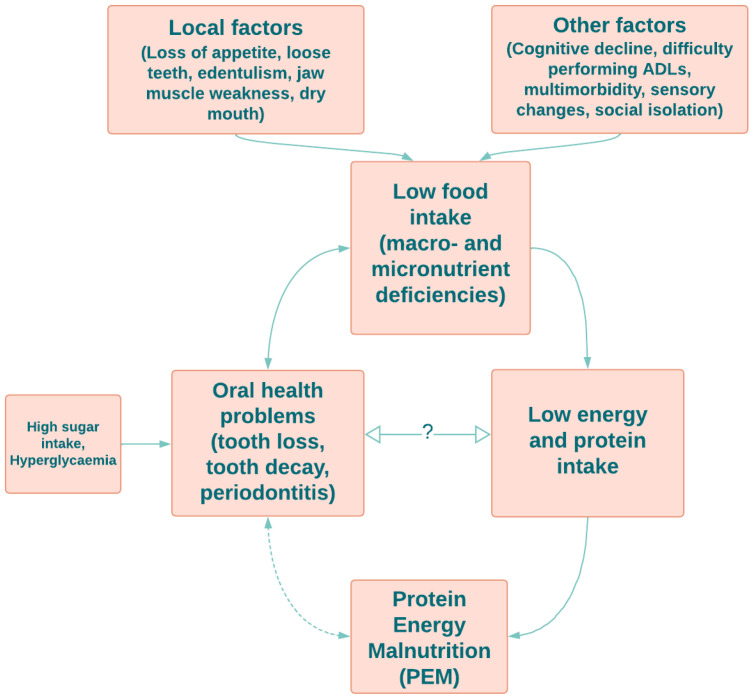
A summary of known and putative links between food and protein intake and oral health in older adults.

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
