# Peer review of "Protein Intake and Oral Health in Older Adults—A Narrative Review"

_nutrients, 2022, doi:10.3390/nu14214478_

Round 1

Reviewer 1 Report

The review shows that there are possible associations between protein intake and oral health in older adults. It is an interesting study. However, there are some issues. The paper needs to be revised.

Though the standards of systematic reviewing cannot be applied to a work, authors should identify evidence through unbiased methods and should document that they have done so, and that they have included all the relevant evidence that they have found.

The authors should clearly state what they add in this review. What is the difference between this review and previous reviews, such as doi: 10.1111/jcpe.12672 or doi: 10.1111/jcpe.12685?

Please add some sentences and more references in the introduction.

Some associations are unclear in the Figure 1; i.e., “Low food intake” to “poor oral health”, “PEM” to “poor oral health” an “poor oral health” to PEM”. Please add appropriate references clearly. The part, “6.1” is not enough. Furthermore, “poor oral health” may not affect “PEM” directly. The authors should consider tooth loss by poor oral health and indirect associations.

Line 175; Please add appropriate references. Are there nutrients to contribute to strength of teeth and jaw bone in older adults?

Line 199; “Tooth mousse” is a product name. Please revise the sentence appropriately or delete it.

Please refer recent papers as below.

doi: 10.3390/ijerph191811354.

doi: 10.1016/S2666-7568(22)00171-4.

doi: 10.1016/j.nutres.2022.08.001.

doi: 10.3290/j.ohpd.b3240807.

doi: 10.1371/journal.pone.0259966.

doi: 10.1093/nutrit/nuab048.

doi: 10.2147/CCIDE.S288137.

There is no reference of “Lanz, 2017” in the reference list.

Please add each material (human, animal, or cell culture) in the Table 1. The current form can lead misunderstanding.

There are some typos; Line 13, Line 112, Line 309, Line 313, etc. Please revise the whole part very carefully.

Please add the limitation of this review.

Based on the reference list, it is very difficult to conclude that protein plays a vital role in oral health in the older adults because there are no sufficient evidences. Please revise the conclusion and abstract appropriately.

Author Response

Response to Reviewers

Dear Reviewer 1,

We appreciate you for your precious time in reviewing our paper and providing valuable comments. It was your valuable and insightful comments that led to possible improvements in the current version. The authors have carefully considered the comments and tried our best to address every one of them. We hope the manuscript after careful revisions meets your high standards of the "Nutrients" journal. The authors welcome further constructive comments if any. Below we provide the point-by-point responses. All modifications in the manuscript have been done with track changes. A clean version of the manuscript has also been uploaded for your reference.

  1. Comment: Though the standards of systematic reviewing cannot be applied to a work, authors should identify evidence through unbiased methods and should document that they have done so, and that they have included all the relevant evidence that they have found.

Response: Authors value the comment of the reviewer and we have now mentioned methods used after introduction section. Methods section now reads as "We performed a search of the literature on PubMed, Web of Science and SciVerse Scopus® databases in June 2022. There were no restrictions on publication dates and the search terms included "older adults or elderly", "malnutrition or low protein intake", "oral or dental health issues". Title and abstract of the articles were reviewed to identify scientific publications relevant to our aim. Specifically, manuscripts that were related to associations between protein intake / malnutrition, and oral health issues in older adults were gathered and reviewed to determine suitability for inclusion".

We also made sure we have included all the relevant literature in the manuscript to best of our knowledge.

  1. Comment: The authors should clearly state what they add in this review. What is the difference between this review and previous reviews, such as doi: 10.1111/jcpe.12672or doi: 10.1111/jcpe.12685?

Response: Authors thank reviewer for the insightful comment. Based on your suggestion, we have included both these articles in the manuscript now. Narrative review - 10.1111/jcpe.12672 explains the roles of macro- and micronutrients in relation to dental caries, gingival bleeding and destructive periodontal disease. Article -10.1111/jcpe.12685? discusses inherited and acquired risk factors for dental caries and periodontal diseases, how periodontal diseases are related to diet and most important dietary risk factors. Our review compiled relevant evidence to specifically show the link between dietary protein and oral health in older adults.

Response: Authors thank the reviewer for the insightful comment.

The narrative review by Hujoel & Lingström - 10.1111/jcpe.12672 is an overview of the roles of macro- and micronutrients in relation to dental caries, gingival bleeding and destructive periodontal disease in the general population. The section covering protein was very brief, no doubt due to the lack of available studies to discuss.

The Consensus article -10.1111/jcpe.12685? was previously unknown to us, and we appreciate the opportunity to refer to this work, particularly regarding inherited risk factors for dental caries and periodontal diseases.

Based on your suggestion, we have referred to both articles in the manuscript.

The main point of difference is that our review is focussed on compiling evidence demonstrating known and plausible links between dietary protein and oral health in older adults.

Thank you again for the opportunity to include these points.

  1. Comment: Please add some sentences and more references in the introduction.

Response: Thank you. The introduction has been expanded, as suggested, with appropriate additional references.

  1. Comment: Some associations are unclear in the Figure 1; i.e., “Low food intake” to “poor oral health”, “PEM” to “poor oral health” an “poor oral health” to PEM”. Please add appropriate references clearly. The part, “6.1” is not enough. Furthermore, “poor oral health” may not affect “PEM” directly. The authors should consider tooth loss by poor oral health and indirect associations.

Response: Authors thank you for the suggestion. Figure 1, legend and 6.1 have been revised based on your comment. We agree that the direct impact of poor oral health on PEM is, as yet, unknown. Loss of teeth due to poor oral health is referred to in our introductory section.

  1. Comment: Line 175; Please add appropriate references. Are there nutrients to contribute to strength of teeth and jaw bone in older adults?

Response: The sentence has been revised as: "Furthermore, other nutrients in protein-rich food (especially animal protein), such as calcium and vitamin D have a beneficial effect on tooth retention in older adults (Krall, Wehler, Garcia, Harris, & Dawson-Hughes, 2001)".

  1. Comment: Line 199; “Tooth mousse” is a product name. Please revise the sentence appropriately or delete it.

Response: Thank you for pointing this out. The study cited studied a trademarked product - we appreciate the opportunity to amend the manuscript. The text now reads: "Additionally, the incorporation of ACP-CPP into oral care products has been shown to prevent formation of biofilm by Streptococcus mutans, a common cariogenic bacterium in the oral cavity involved in plaque formation (Sionov et al., 2021)".

  1. Comment: Please refer recent papers as Sentence was revised based on your suggestion. It now reads as "CPP-ACP prevents biofilm formation of Streptococcus mutans, a common cariogenic bacterium in the oral cavity involved in plaque formation thereby reducing the risk for dental caries (Sionov et al., 2021)".
  2. Comment: Please refer recent papers as below.

doi: 10.3390/ijerph191811354.

Yeung (2022): A Nutritionally Complete Oral Nutritional Supplement Powder Improved Nutritional Outcomes in Free-Living Adults at Risk of Malnutrition: A Randomized Controlled Trial

doi: 10.1016/S2666-7568(22)00171-4.

Thomson 2022: Effectiveness and cost-effectiveness of oral nutritional supplements in frail older people who are malnourished or at risk of malnutrition: a systematic review and meta-analysis

doi: 10.1016/j.nutres.2022.08.001.

Kioka (2022) Soy protein intake increased bone mineral density under nonenergy-deficiency conditions but decreased it under energy-deficiency conditions in young female rats

doi: 10.3290/j.ohpd.b3240807.

Hwang (2022) The Relationship Between Periodontal Disease and Nutrient Intake in Korean Adults: The Korea National Health and Nutrition Examination Survey (KNHANES VII) from 2016-2018

doi: 10.1371/journal.pone.0259966.

Haruyama (2021) Leucine rich amelogenin peptide prevents ovariectomy-induced bone loss in mice

doi: 10.1093/nutrit/nuab048.

Choowong (2022) Macronutrient-induced modulation of periodontitis in rodents-a systematic review

doi: 10.2147/CCIDE.S288137.

Santonocito (2021) Dietary Factors Affecting the Prevalence and Impact of Periodontal Disease [Review]

Response: We thank the reviewer for suggesting these recent articles in the area of nutrition and oral health. They have been cited in the manuscript where necessary.

  1. Comment: There is no reference of “Lanz, 2017” in the reference list.

Response: We apologise for the mistake. Lanz, 2017 has been added to the reference list as below:

Lanz, T. V., Becker, S., Mohapatra, S. R., Opitz, C. A., Wick, W., & Platten, M. (2017). Suppression of Th1 differentiation by tryptophan supplementation in vivo. Amino Acids, 49(7), 1169–1175. https://doi.org/10.1007/s00726-017-2415-4

  1. Comment: Please add each material (human, animal, or cell culture) in the Table 1. The current form can lead misunderstanding.

Response: Thank you for the valuable suggestion. A column has been added to the Table 1 including each material as human, animal or cell culture.

  1. Comment: There are some typos; Line 13, Line 112, Line 309, Line 313, etc. Please revise the whole part very carefully.

Response: Typos in Line 13, Line 112, Line 309 and Line 313, and others in the manuscript were corrected.

  1. Comment: Please add the limitation of this review. 

Response: Thank you for your suggestion. We have included limitations of the review and it now reads as "As in any narrative review, we had no explicit criteria for article selection. We did not conduct a formal assessment of the quality or risk of bias of included studies, which can be considered a limitation of this review. The authors acknowledge that a systematic review, as a gold standard summary of evidence is best to guide clinical decision-making and inform policy development. However, there is insufficient evidence from interventional and cohort studies in this area to inform such a review, or to identify cause-and-effect relationships. Bringing attention to the need for investigations related to oral health and protein intake in older adults is an important step in raising awareness of the need for more evidence on this topic".

  1. Comment: Based on the reference list, it is very difficult to conclude that protein plays a vital role in oral health in the older adults because there are no sufficient evidences. Please revise the conclusion and abstract appropriately.

Response: Thank you for your insightful comment. We agree that at this time, there is scant evidence, and as per the introduction, this review focusses on “theoretical evidence” of the oral health-protein relationship. We firmly believe this is a significant gap in the literature.

Our conclusion now reads: "Older adults experience both oral health issues and malnutrition, which have a coadjutant effect on each other. Evidence suggests that the effect of protein intake on oral health may be mediated through malnutrition. There is a tendency for lower protein intake among older adults yet the impact of this on oral health is unexplored. Examination of the impact of inadequate protein intake on oral health in this demographic is required. Evaluation of associations would inform guidelines for optimal protein requirements for ensuring of good oral health in older adults".

Thank you.

Sincerely,

Thilini N Jayasinghe, PhD

Sydney School of Dentistry and Charles Perkins Centre

The University of Sydney

Australia

Reviewer 2 Report

Dear Authors,

nice and readable narrative review. I learned a lot!

Only few minor corrections:

-Introduction, line 29: write oral cancer instead of "oral cancers"

-correct double space between the words:

abstract, line 13 Evidence;introduction, line 39 macroglobulin.

Best regards

Author Response

Dear Reviewer 2,

We appreciate your precious time in reviewing our paper and providing valuable comments. It was your valuable and insightful comments that led to possible improvements in the current version. The authors have carefully considered the comments and tried our best to address every one of them. We hope the manuscript, after careful revisions, meets your high standards of the "Nutrients" journal. The authors welcome further constructive comments, if any. Below we provide the point-by-point responses. All modifications in the manuscript have been done with track changes. A clean version of the manuscript has also been uploaded for your reference.

  1. Comment: Introduction, line 29: write oral cancer instead of "oral cancers"

Response: Thank you for pointing this out. We have changed "oral cancers" to "oral cancer" based on your comment.

  1. Comment: correct double space between the words: abstract, line 13 Evidence; introduction, line 39 macroglobulin.

Response: Thank you. We have now rechecked the manuscript and changed the double space to single space.

Thank you.

Sincerely,

Thilini N Jayasinghe, PhD

Sydney School of Dentistry and Charles Perkins Centre

The University of Sydney

Australia

Round 2

Reviewer 1 Report

The revision was acceptable.